

# Social and environmental factors modulate leucocyte profiles in free-living Greylag geese (*Anser anser*)

Didone Frigerio[1,2], Sonja C. Ludwig[1,3], Josef Hemetsberger[1,2], Kurt Kotrschal[1,2] and Claudia A.F. Wascher[1,4]

[1] Core Facility Konrad Lorenz Forschungsstelle for Behaviour and Cognition, University of Vienna, Grünau im Almtal, Austria
[2] Department of Behavioural Biology, University of Vienna, Vienna, Austria
[3] Game & Wildlife Conservation Trust, The Coach House, Eggleston Hall, Barnard Castle, United Kingdom
[4] Department of Life Sciences, Anglia Ruskin University, Cambridge, United Kingdom

## ABSTRACT

**Background**. Blood parameters such as haematocrit or leucocyte counts are indicators of immune status and health, which can be affected, in a complex way, by exogenous as well as endogenous factors. Additionally, social context is known to be among the most potent stressors in group living individuals, therefore potentially influencing haematological parameters. However, with few exceptions, this potential causal relationship received only moderate scientific attention.

**Methods**. In a free-living and individually marked population of the highly social and long-lived Greylag goose, *Anser anser*, we relate variation in haematocrit (HCT), heterophils to lymphocytes ratio (H/L) and blood leucocyte counts to the following factors: intrinsic (sex, age, raising condition, i.e. goose- or hand-raised), social (pair-bond status, pair-bond duration and parental experience) and environmental (biologically relevant periods, ambient temperature) factors. Blood samples were collected repeatedly from a total of 105 focal birds during three biologically relevant seasons (winter flock, mating season, summer).

**Results**. We found significant relationships between haematological parameters and social as well as environmental factors. During the mating season, unpaired individuals had higher HCT compared to paired and family individuals and this pattern reversed in fall. Similarly, H/L ratio was positively related to pair-bond status in a seasonally dependent way, with highest values during mating and successful pairs had higher H/L ratio than unsuccessful ones. Also, absolute number of leucocytes tended to vary depending on raising condition in a seasonally dependent way.

**Discussion**. Haematology bears a great potential in ecological and behavioural studies on wild vertebrates. In sum, we found that HTC, H/L ratio and absolute number of leucocytes are modulated by social factors and conclude that they may be considered valid indicators of individual stress load.

Corresponding author
Didone Frigerio,
didone.frigerio@univie.ac.at

## INTRODUCTION

Haematocrit (HCT) and differential leucocyte count are important diagnostic tools in gaining information about an animal's condition and health and may be regarded as indicators of individual responses to environmental and social conditions (*Cooper, 1975*; *Gavett & Wakeley, 1986*; *Hellgren, Vaughan & Kirkpatrick, 1989*; *Averbeck, 1992*; *Saino et al., 1997*; *Ots, Murumägi & Hõrak, 1998*; *Bortolotti et al., 2009*; *Vinkler et al., 2010*).

In wild birds HCT, i.e., the relative volume of red blood cells compared to the total blood volume (*Harrison & Harrison, 1986*), has been used to indicate the animals' physical condition (e.g., *Saino et al., 1997*; *Ots, Murumägi & Hõrak, 1998*) and is known to decrease in response to stressful conditions (*Dickens, Earle & Romero, 2009*). However, HCT also varies with sex, age, reproductive status, as well as geographic distribution (*Rehder et al., 1982*; *Rehder, Bird & Laguë, 1982*; *Dawson & Bortolotti, 1997a.*; *Dawson & Bortolotti, 1997b*; *Bearhop et al., 1999*; *Potti et al., 1999*; *Fair, Whitaker & Pearson, 2007*).

The differential leucocyte count may be regarded as a proper proxy of individual immune function (*Dufva & Allander, 1995*; *Zuk, Johnsen & MacLarty, 1995*; *Johnsen & Zuk, 1998*), providing information on the relative occurrence of different leucocyte types. Absolute leucocyte numbers generally scale negatively with body condition (as an indicator of health, see *Verhulst, Oosterbeek & Bruinzeel, 2002*). In birds, heterophils are rather indicative of changes in the environment (*Gross & Siegel, 1983*; *Maxwell & Robertson, 1998*). As lymphocyte numbers decrease while heterophil numbers increase in response to stressful conditions, the ratio of heterophils/lymphocytes (H/L ratio) is used as an indicator of physiological stress (e.g., *Gross & Siegel, 1983*; *Gross & Siegel, 1986*; *McFarlane & Curtis, 1989*; *Maxwell, 1993*; *Vleck et al., 2000*; *Lebigre et al., 2012*). Finally, monocytes are long-lived phagocytic cells associated, together with other leukocyte types (e.g., eosinophils, basophils), with defence against infections and bacteria (*Davis, Maney & Maerz, 2008*).

Social interactions are known to be among the most potent stressors in group living individuals (*Von Holst, 1998*; *De Vries, Glasper & Detillion, 2003*). Therefore, it may also influence haematological parameters. However, with the exception of a few veterinary studies (e.g., *Arfuso et al., 2016*; *Zakari et al., 2016*), this potential causal relationship (i.e., social context and haematology) received only moderate scientific attention. In Black-capped Chickadees (*Poecile atricapillus*), for example, dominant males had higher haematocrits compared to subordinates (*Van Oort et al., 2007*) and in yellow baboons (*Papio cynocephalus*) frequent involvement in aggressive encounters was related with decreased lymphocyte count (*Alberts, Sapolsky & Altmann, 1992*). On the other side, several studies suggest a relationship between steroid hormones and haematocrit (e.g., a positive one with testosterone in White-plumed Honeyeaters *Lichenostomus penicillatus* and American Kestrels *Falco sparverius*, *Rehder, Bird & Laguë, 1982*; a negative one with corticosterone in both sexes of two species of longspurs, *Calcarius ornatus* and *C.mccownii*, *Lynn, Hunt & Wingfield, 2003*; for a review, see *Fair, Whitaker & Pearson, 2007*), even though the exact mechanisms remain unclear.

We investigated variation in haematocrit and blood leucocytes in relation to endogenous (age, sex) and exogenous factors (social status, season) in a free living and

individually marked population of the socially complex greylag geese, *Anser anser* (*Lorenz, 1988*; *Kotrschal, Hemetsberger & Weiß, 2006*; *Kotrschal, Scheiber & Hirschenhauser, 2010*; *Hemetsberger, Scheiber & Weiß, 2013*). Social context is known to be among the strongest modulators of the physiological stress responses in greylag geese (e.g., *Wascher, Scheiber & Kotrschal, 2008*; *Wascher, Arnold & Kotrschal, 2008*; *Wascher et al., 2009*; *Kralj-Fiser et al., 2010*), which in turn are alleviated via emotional social support by partners (*Frigerio, et al., 2003*; *Scheiber et al., 2005*; *Scheiber, Kotrschal & Weiß, 2009*; *Wascher et al., 2012*). Across seasons, males and females are faced with different demands (*Kotrschal, Scheiber & Hirschenhauser, 2010*); consequently, physiological changes, such as levels of corticosterone, co-vary with seasonal variation in behaviour (*Hirschenhauser, Moestl & Kotrschal, 1999a*; *Hirschenhauser, Moestl & Kotrschal, 1999b*; *Frigerio et al., 2004a*). The reproductive season of greylag geese starts in January. Testosterone and corticosterone levels increase over the season (*Kotrschal, Hirschenhauser & Moestl, 1998*; *Hirschenhauser, Moestl & Kotrschal, 1999a*), as do agonistic encounters between individuals (*Lorenz, 1988*). With the beginning of the breeding season (March to July) the flock disintegrates into pairs. As for most bird species, reproduction in geese is energetically costly, especially for the females who lay and incubate the eggs (*Raveling, 1979*; *Thompson & Raveling, 1988*). Parental geese show elevated corticosterone levels (*Kotrschal, Hirschenhauser & Moestl, 1998*) and synchronise their moult with the rearing period, so that they start flying again together with their offspring, when the goslings are approximately 10 weeks old (*Lorenz, 1988*). In alignment with the parental phase (i.e., from hatching to fledging), androgen levels reach their annual minimum in both sexes. The flock re-unites in August after moulting, when androgens reach a second peak in unpaired males, while in paired individuals the second peak is reached later in fall (*Hirschenhauser, Moestl & Kotrschal, 1999a*; *Hirschenhauser, Moestl & Kotrschal, 1999b*). This hints at the close interplay between physiology and social status in greylag geese, as suggested by further studies (e.g., stress response and social allies: *Sachser, Duerschlag & Hirzel, 1998*; heart rate in the context of sociality: *Aureli, Preston & De Waal, 1999*; steroid hormones and social status: *Wingfield, Hegner & Lewis, 1991*; *Goymann, Villavicencio & Apfelbeck, 2015*).

We studied seasonal and individual variation in HCT and leucocyte counts. We tested whether (a) individual factors, such as sex, age or raising condition, (b) social factors, such as pair-bond status, (c) biologically relevant periods (mating season, after moult, autumn), or a combination of the above, accounted for variation in haematological parameters. We hypothesized that individuals of different sexes and social status (females and males, paired versus unpaired) are faced with different social and energetic demands within the flock (e.g., *Metcalfe, Taylor & Thorpe, 1995*; *Senar et al., 2000*), which in turn, will affect the immune system and will be manifest in HCT and leucocyte count. Since the mating season is socially and energetically more challenging than moult and the stable winter flock (*Drent & Daan, 1980*) we predict that both haematological measurements will peak during this period, particularly in the high-ranking parental individuals. Furthermore, we expect to find seasonal differences between males and females, contingent with behavioural and physiological differences between the sexes depending on an individual's social environment (*Wingfield, Hegner & Lewis, 1991*; *Kotrschal, Hirschenhauser & Moestl, 1998*;

*Hirschenhauser, Moestl & Kotrschal, 1999a; Hirschenhauser, Moestl & Kotrschal, 1999b; Owens & Hartley, 1998; Wascher et al., 2012; Lees et al., 2012*). Additionally, as hand-raised individuals generally differ from parent-raised conspecifics in glucocorticoid stress reactivity (*Hemetsberger et al., 2010*), we finally predict that hand-raised geese will show higher values of both haematological parameters during socially challenging periods than parent-raised ones.

## MATERIAL AND METHODS

### Ethical statement

This study complies with all current Austrian laws and regulations concerning the work with wildlife. Catching and blood sampling of focal individuals was permitted under Animal Experiment License Nr. 66.006/0010-II/10b/2010 by the Austrian Federal Ministry for Science and Research.

### Study area and focal animals

The Konrad Lorenz Research Station (KLF) is situated at 550 m above sea level in the valley of the river Alm in the Northern part of the Austrian Alps (47°48′E, 13°56′N). The non-migratory flock of greylag geese we studied was introduced by Konrad Lorenz and co-workers in 1973 (*Lorenz, 1988*). The geese are unrestrained and generally spend the day close to the research station where they are provided with supplemental food (pellets and grain) twice a day year-round. At night, the birds roost on a lake approximately 10 km to the South (Almsee). As in wild populations, natural predators (mainly red foxes, *Vulpes vulpes* and golden eagles, *Aquila chrysaetos*) may account for losses up to 10 % of the flock per year (*Hemetsberger, 2001; Hemetsberger, 2002*). All geese are marked with coloured leg rings and are habituated to the close presence of humans. They neither show increased immunoreactive corticosterone metabolites in their droppings nor heart rate increases when approached by familiar humans (*Scheiber et al., 2005; Wascher et al., 2012*). Social behaviour and individual life-history data have been monitored since 1973. Such long-term observations provide reliable information about an individual's social relationships within the flock (i.e., paired or not). Approximately 20% of the flock members are carefully hand-raised in the frame of specific research projects. Details about the hand-raising tradition of the KLF are published elsewhere (*Hemetsberger, Scheiber & Weiß, 2013*). During the period of data collection (autumn 2010 to spring 2013) the number of geese in flock varied between 153 and 157 individuals.

Focal animals of the present study were 105 greylag geese of different age (at the time of sampling ranging from 0.16 to 22.81 years, mean age $\pm$ SE $= 5.06 \pm 0.36$), sex (67 males, 49 females), social status (68 paired individuals, 30 family individuals, 18 unpaired ones,) and different raising condition (71 goose-raised and 45 hand-raised individuals).

### Data collection

Data were collected during three phases, representing biologically relevant periods of the year (*Hemetsberger, Scheiber & Weiß, 2013*): (1) in summer, after moulting; (2) in autumn (winter flock) and (3) during the mating season before the beginning of the laying period

(approx. mid-March). Focal geese were caught by hand up to three times, once in each period either by a familiar human observer who approached the geese from the back and picked them up when being close enough or in a 'trapping enclosure,' which the geese entered voluntarily during feeding. In this way, chases or strong wing flapping were avoided. To control for diurnal variation of the focal parameters, all geese were sampled in the morning between 0730 and 0930.

Weather data were provided by a weather station in Grünau (47°51′E, 13°57′N) operated by Max Rauscher (http://www.gruenau.tv; last accessed 25 December 2014). Temperature data were recorded every 5 min, for which we calculated daily means during our analysis.

## Blood samples

A total number of 169 blood samples ($N = 105$ individuals, Table 1) were collected by puncturing the tarsal vein with a sterile needle (24 μm diameter) and collecting blood in two heparinized micro-haematocrit capillaries (75 mm). Furthermore, we measured tarsus, beak and wing length, and body weight. The whole procedure lasted less than 7 min per individual.

In order to determine an individual's blood cell count (*Prinzinger, Misovic & Nagel, 2012*) one drop of blood was smeared onto a microscope slide, air-dried and stored until later identification of leucocytes at the Clinical Pathology Platform of the University of Veterinary Medicine in Vienna (Austria). Differential blood cells count provided information on absolute leucocyte number/μl (LEUCO) and the relative occurrence of different leucocyte types (heterophils, lymphocytes, monocytes, basophils and eosinophils, *Prinzinger, Misovic & Nagel, 2012*). Blood smears were Romanowsky-stained with Haemaquick (E. Lehmann GmbH, Salzburg Austria) and microscopically evaluated. Thereafter, 100 white blood cells were differentiated into heterophilic, eosinophilic or basophilic granulocytes, monocytes, and lymphocytes at 1,000× magnification using oil immersion. Results were provided in percentages.

The haematocrit capillaries were then sealed with plasticine at the bottom and centrifuged at 8,000 rpm for 5 min in order to determine the HCT. Volumes of red blood cell and plasma respectively were measured on the capillaries to the nearest 0.5 mm with calipers. HCT was then calculated as ratio as follow (*Prinzinger, Misovic & Nagel, 2012*): red blood cell volume/(red blood cell volume + plasma volume). The individual arithmetic mean of the two HCT values was then taken into further analyses.

## Statistical analysis

We ran linear mixed effect models (LMMs) fitted with maximum likelihood using the nlme package *Pinheiro et al., 2015*); model formulation and computational methods are described in *Lindstrom & Bates, 1990*) in R (*R Core Team, 2015*). Haematocrit, heterophils to lymphocytes (H/L) ratio, leucocyte numbers and percentage of monocytes were considered each in turn as response variable and four LMMs were calculated for each of the three data subsets. Subset (1) included all 169 samples collected from 105 individuals. We tested 7 competing hypotheses: (a) individual factors accounting for variation in haematological parameters with sex, age (i.e., days of life) and raising condition (hand-raised, goose-raised) included as fixed factors; (b) social factors (pair-bond status: unpaired,

 

Frigerio et al. (2017), *PeerJ*, DOI 10.7717/peerj.2792

**Table 1  Details about the number of sampled individuals and the number of collected samples per season and category (i.e., sex, social status, rearing condition).**

| Season | Nr. of samples | Nr. of sampled individuals | Nr. of sampled individuals by sex | | Nr. of samples by sex | Nr. of sampled individuals by rearing condition | | Nr. of samples by rear. cond. | Nr of sampled individuals by social status | | Nr. of samples by soc. stat. |
|---|---|---|---|---|---|---|---|---|---|---|---|
| After Molt | **23** | **23** | Males | 10 | 10 | Goose-raised | 23 | 23 | Single | 0 | 0 |
| | | | Females | 13 | 13 | Hand-raised | 0 | 0 | Paired | 0 | 0 |
| | | | | | | | | | Family | 23 | 23 |
| Autumn | **82** | **73** | Males | 37 | 40 | Goose-raised | 37 | 37 | Single | 8 | 8 |
| | | | Females | 36 | 42 | Hand-raised | 36 | 45 | Paired | 39 | 48 |
| | | | | | | | | | Family | 26 | 26 |
| Mating | **64** | **61** | Males | 40 | 42 | Goose-raised | 39 | 39 | Single | 16 | 18 |
| | | | Females | 21 | 22 | Hand-raised | 22 | 25 | Paired | 30 | 31 |
| | | | | | | | | | Family | 15 | 15 |

paired, family) explaining variation in haematological parameters; (c) biologically relevant period (mating season, after moult, autumn) accounting for variation in haematological parameters; (d) Social factors (pair-bond status) explaining variation in a seasonally dependent way (the interaction between season and pair-bond status was additionally included in the model); (e) individual factors (sex, age, raising-condition), season and the interactions between season and individuals were included in the model; (f) individual factors, pair-bond status and the interaction between individual factors and pair-bond status; (g) null model. Subset (2) included only data from male–female pair bonded individuals (68 samples from 50 individuals). We tested three hypotheses: (a) pair-bond duration, defined as 'established' if the pair spent already one breeding season together or 'newly', if this was not the case or (b) parental experience, i.e., if the pair previously successfully raised fledged offspring (yes/no), accounting for variation in haematological parameters; (c) null model.

The third subset of data only included individuals from which morphological measurements were collected besides blood samples (116 samples from 70 individuals). A model including the body size index (BSI), calculated as ratio weight/tarsus (as indicator for body's structural size, *sensu Green, 2001*) as fixed factor, was tested against the null model. In order to account for repeated measures for each observed individual, the individual identity was included as random factor. We based our model selection on corrected Akaike's Information Criterion values (AICc). We calculated the difference between the best model and each other possible model (ΔAICc) and ranked the model combinations according to their ΔAICc, which provides an evaluation of the overall strength of each model in the candidate set. Different competing models tested are presented in Table 2. If multiple models qualified as the similarly good models, i.e., ΔAICc ≤ 2 (*Burnham & Anderson, 2002*; *Burnham, 2004*) we applied a model averaging approach, which calculates model averaged parameters using the MuMIn package (version 1.15.6). In both data subset (2) pair-bonded individuals and (3) morphological measures, for some of the parameters the candidate models did not qualify as better as compared to the null model (Table 2) and are therefore not presented in the results section.

## RESULTS

(1) Full dataset:

The best candidate model explaining variation in haematocrit included individual factors (sex, age, raising condition), season as well as the interactions between season and individual factors. HCT significantly increased with age in autumn but not during the mating season or after moult (Table 3). The opposite pattern was found for percentage of monocytes, which tended to decrease with age in autumn. The H/L ratio varied depending on pair-bond status in a seasonally dependent way (Fig. 1).

Leucocyte counts differed depending on pair-bond status in a seasonally dependent way. Generally, leucocyte counts were highest after moult (family, mean ± SD: 11086.956 ± 4494.067) and in autumn (unpaired: 13125 ± 6300.51; paired: 13175 ± 4948.741; family: 9519.23 ± 3548.184) and decreased during the mating season (unpaired: 7933.333

**Table 2** **Model selection of analyses examining factors affecting (a) haematocrit (HCT), (b) haeterophily to lymphocytes ratio (H/L ratio), (c) leucocyte count and (d) percentage of monocytes.** Individual identity was fitted as a random term. Three subsets of data have been tested: (1) the full-dataset included all 169 samples collected from 105 individuals; subset (2) included data from male–female pair bonded individuals (68 samples from 50 individuals); subset (3) included only data from individuals from which morphological measurements were collected (116 samples from 70 individuals). LogLik, log-likelihood; AICc, second order Akaike's Information Criterion; ΔAICc, difference between the best model and each other possible model.

| Models | df | LogLik | AIC | ΔAICc |
|---|---|---|---|---|
| **(1a) Full-dataset: HCT** | | | | |
| Sex, age, raising history, season, season*sex, season*age, season*raising-status | 10 | 242.574 | −463.757 | 0 |
| Pair-bond status, season, season*pair-bond status | 6 | 233.649 | −454.78 | 8.976 |
| Season | 4 | 230.983 | −453.722 | 10.034 |
| Sex, age, raising history | 6 | 219.471 | −426.423 | 37.333 |
| Pair-bond status | 4 | 215.376 | −422.508 | 41.248 |
| Sex, age, raising history, pair-bond status, pair-bond status*sex, pair-bond status*age, pair-bond status*raising status | 8 | 218.177 | −419.454 | 44.302 |
| Null model | 3 | 207.589 | −409.032 | 54.724 |
| **(1b) Full-dataset: H/L ratio** | | | | |
| Pair-bond status, season, season*pair-bond status | 6 | −196.356 | 405.23 | 0 |
| Sex, age, raising history, season, season*sex, season*age, season*raising-status | 10 | −198.064 | 417.521 | 12.29 |
| Season | 4 | −205.476 | 419.196 | 13.965 |
| Sex, age, raising history | 6 | −207.444 | 427.406 | 22.176 |
| Sex, age, raising history, pair-bond status, pair-bond status*sex, pair-bond status*age, pair-bond status*raising status | 8 | −205.462 | 427.824 | 22.593 |
| Pair-bond status | 4 | −209.898 | 428.04 | 22.81 |
| Null model | 3 | −213.424 | 432.994 | 27.763 |
| **(1c) Full-dataset: leucocyte count** | | | | |
| Pair-bond status, season, season*pair-bond status | 6 | −1650.015 | 3312.549 | 0 |
| Sex, age, raising history, season, season*sex, season*age, season*raising-status | 10 | −1646.151 | 3313.715 | 1.166 |
| Season | 4 | −1656.422 | 3321.087 | 8.538 |
| Null model | 3 | −1660.982 | 3328.11 | 15.56 |
| Sex, age, raising history | 6 | −1658.534 | 3329.586 | 17.036 |
| Pair-bond status | 4 | −1660.977 | 3330.199 | 17.649 |
| Sex, age, raising history, pair-bond status, pair-bond status*sex, pair-bond status*age, pair-bond status*raising status | 8 | −1657.008 | 3330.917 | 18.367 |
| **(1d) Full-dataset: percentage of monocytes** | | | | |
| Sex, age, raising history, season, season*sex, season*age, season*raising-status | 10 | −480.402 | 982.197 | 0 |
| Pair-bond status, season, season*pair-bond status | 6 | −485.006 | 982.531 | 0.334 |
| Season | 4 | −491.375 | 990.995 | 8.797 |
| Sex, age, raising history | 6 | −491.792 | 996.104 | 13.906 |
**Table 2** (*continued*)

| Models | df | LogLik | AIC | ΔAICc |
|---|---|---|---|---|
| Pair-bond status | 4 | −494.597 | 997.439 | 15.241 |
| Null model | 3 | −496.611 | 999.369 | 17.171 |
| Sex, age, raising history, pair-bond status, pair-bond status*sex, pair-bond status*age, pair-bond status*raising status | 8 | −493.017 | 1002.935 | 20.737 |
| **(2a) Pair-bonded individuals: HCT** | | | | |
| Null model | 3 | 90.11 | −173.846 | 0 |
| Pair-bond duration | 4 | 90.84 | −173.046 | 0.8 |
| Parental experience | 4 | 90.276 | −171.918 | 1.928 |
| **(2b) Pair-bonded individuals: H/L ratio** | | | | |
| Parental experience | 4 | −55.067 | 118.769 | 0 |
| Pair-bond duration | 4 | −55.911 | 120.458 | 1.688 |
| Null model | 3 | −58.245 | 122.865 | 4.095 |
| **(2c) Pair-bonded individuals: Leucocytes** | | | | |
| Null model | 3 | −665.648 | 1337.672 | 0 |
| Parental experience | 4 | −665.424 | 1339.484 | 1.812 |
| Pair-bond duration | 4 | −665.49 | 1339.616 | 1.944 |
| **(2d) Pair-bonded individuals: Monocytes** | | | | |
| Null model | 3 | −191.664 | 389.704 | 0 |
| Pair-bond duration | 4 | −191.02 | 390.676 | 0.972 |
| Parental experience | 4 | −191.293 | 391.222 | 1.518 |
| **(3a) Morphological measurements: HCT** | | | | |
| BSI | 4 | 175.545 | −342.731 | 0 |
| Null model | 3 | 146.71 | −278.206 | 55.525 |
| **(3b) Morphological measurements: H/L ratio** | | | | |
| BSI | 4 | −146.399 | 301.158 | 0 |
| Null model | 3 | −148.494 | 303.203 | 2.045 |
| **(3c) Morphological measurements: Leucocytes** | | | | |
| Null model | 3 | −1142.87 | 2291.954 | 0 |
| BSI | 4 | −1142.073 | 2292.506 | 0.552 |
| **(3d) Morphological measurements: Monocytes** | | | | |
| BSI | 4 | −345.286 | 698.932 | 0 |
| Null model | 3 | −349.29 | 704.794 | 5.862 |

± 2651.303; paired: 10500 ± 5154.748; family: 9,200 ± 2111.194). This decrease was most pronounced in unpaired individuals as compared to paired and family individuals. Further, leucocyte counts tended to vary seasonally, depending on individual raising condition. In the winter leucocyte counts were higher in hand-raised as compared to goose raised individuals (hand-raised: 13722.222 ± 5248.24; goose-raised: 10,050 ± 3645.861) and this pattern reversed during the mating season (hand-raised: 9468.75 ± 4870.810; goose-raised: 9715.909 ± 3901.993). In summer (after moult) only hand-raised individuals were sampled (11086.956 ± 4494.067).

(2) Pair-bonded individuals dataset

**Table 3 Results of the full statistical models.** Models for each response variable of all three data subsets (1 = full dataset, 2 = paired individuals, 3 = body measures) are presented. Factors in bold are significant ($p \leq 0.05$).

| | Response variable | Estimate ± SE | t value | p |
|---|---|---|---|---|
| **(1) Haematocrit** | | | | |
| | Raising history | −0.078 ± 0.049 | −1.574 | 0.118 |
| | Sex | 0.021 ± 0.032 | 0.669 | 0.504 |
| | Age | 0 ± 0 | 3.815 | <0.001 |
| | Season | 0.656 ± 0.011 | 5.715 | <0.001 |
| | Season*raising history | 0.03 ± 0.019 | 1.564 | 0.123 |
| | Season*sex | −0.02 ± 0.014 | −1.425 | 0.159 |
| | **Season*age** | **0 ± 0** | **−3.572** | **<0.001** |
| **(1) H/L ratio** | | | | |
| | Pair-bond status | 1.245 ± 0.346 | 3.588 | <0.001 |
| | **Season** | **1.192 ± 0.23** | **5.177** | **<0.001** |
| | **Season*pair-bond status** | **−0.563 ± 0.135** | **−4.528** | **<0.001** |
| **(1) Leucocytes** | | | | |
| | **Pair-bond status** | **−1.052 ± 0.293** | **3.545** | **<0.001** |
| | Season | −0.555 ± 0.418 | 1.32 | 0.186 |
| | **Raising history** | **0.801 ± 0.384** | **2.057** | **0.039** |
| | Sex | 0.087 ± 0.253 | 0.339 | 0.734 |
| | Age | 0.687 ± 0.409 | 1.644 | 0.1 |
| | **Season*pair-bond status** | **0.868 ± 0.265** | **3.212** | **0.001** |
| | Season*raising history | −0.689 ± 0.372 | 1.812 | 0.069 |
| | Season*sex | −0.161 ± 0.252 | 0.623 | 0.532 |
| | Season*age | −0.691 ± 0.416 | 1.624 | 0.104 |
| **(1) Monocytes** | | | | |
| | Raising history | −0.645 ± 0.375 | 1.698 | 0.089 |
| | Sex | 0.053 ± 0.249 | 0.212 | 0.831 |
| | Age | 0.516 ± 0.405 | 1.247 | 0.212 |
| | Season | 0.374 ± 0.154 | 2.369 | 0.017 |
| | Pair-bond status | 0.319 ± 0.29 | 1.085 | 0.277 |
| | Season*raising history | 0.471 ± 0.362 | 1.276 | 0.201 |
| | Season*sex | −0.014 ± 0.246 | 0.056 | 0.955 |
| | Season*age | −0.807 ± 0.41 | 1.924 | 0.054 |
| | Season*pair-bond status | −0.004 ± 0.261 | 0.016 | 0.986 |
| **(2) H/L ratio** | | | | |
| | **Pair-bond duration** | **0.268 ± 0.123** | **2.111** | **0.034** |
| | **Parental experience** | **0.315 ± 0.123** | **2.493** | **0.012** |
| **(3) HCT** | | | | |
| | **BSI** | **0 ± 0** | **10.281** | **<0.001** |
| **(3) H/L ratio** | | | | |
| | **BSI** | **0.003 ± 0.001** | **2.048** | **0.046** |
| **(3) Monocytes** | | | | |
| | **BSI** | **0.026 ± 0.008** | **3.248** | **0.002** |

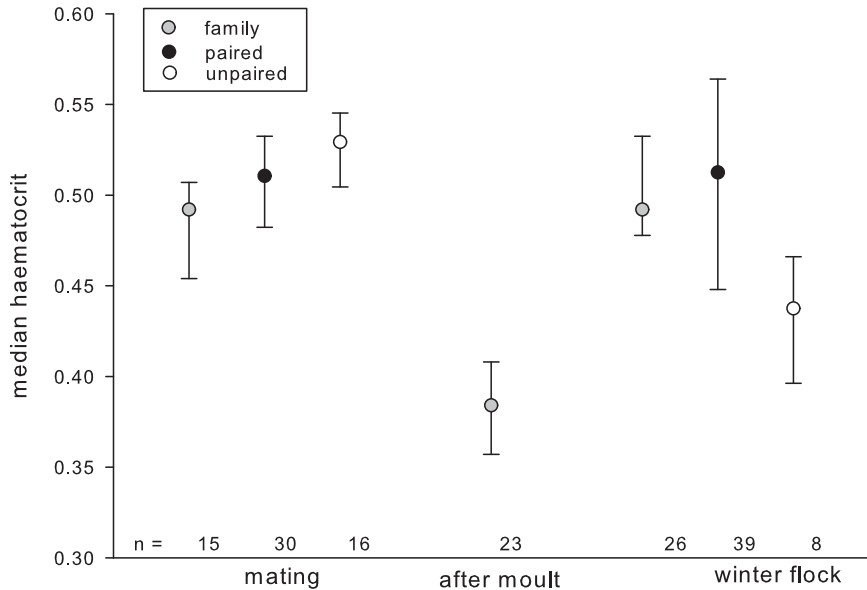

**Figure 1 Haematocrit in relation to season and pair-bond status.** Gray circles represent individuals with offspring (i.e., family); black circles represent paired individuals without offspring; plain circles represent unpaired individuals. Circles indicate median, error bars represent interquartile ranges (lower: 25th and upper: 75th percentile).

HCT, leucocyte number, and percentage of monocytes were not significantly affected by pair-bond duration or parental experience. H/L ratios were higher in pairs with an 'established' pair-bond duration compared to those with a 'newly' formed one (established: $1.228 \pm 0.593$; newly: $0.819 \pm 0.418$) and were higher in reproductively successful as compared to unsuccessful pairs (Fig. 2).

(3) Morphological measures

The body size index (BSI) was positively related to HCT, percentage of monocytes and H/L ratio.

## DISCUSSION

Our results show that social and environmental factors interact with individual physiology in a complex way. We found that in the free-living greylag geese investigated, haematocrit (HCT) and differential leucocyte counts are seemingly contingent with a suite of endogenous (i.e., sex, age, raising condition), social (i.e., pair-bond status, pair-bond duration and parental experience) and environmental factors (i.e., biologically relevant seasons).

Interestingly, pair-bond status showed a seasonal dependent relationship with several haematological parameters: unpaired individuals had the highest HCT during the mating season and H/L ratio was significantly higher in individuals who successfully raised young as compared to individuals which failed to fledge offspring that year. Our results hint at a complex relationship between an individual's social status within the flock, the seasonal patterns of corticosterone and individual behavioural investment in order to optimize the own fitness, as discussed by *Kotrschal, Hirschenhauser & Moestl (1998)*. In fact, during

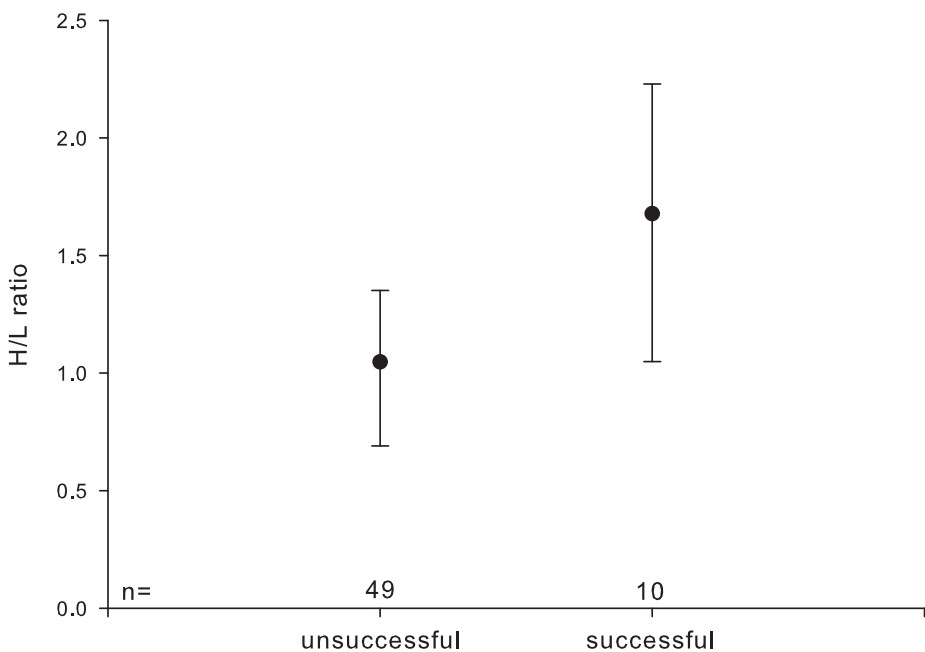

**Figure 2** **Heterophil to lymphocyte (H/L) ratio in relation to individuals' reproductive success.** Circles indicate median, error bars represent interquartile ranges (lower: 25th and upper: 75th percentile).

summer and winter, the paired males with offspring had significantly higher corticosterone than both paired males without offspring and singletons, whereas during the mating season, singletons had marginally higher corticosterone than paired males. Furthermore, stress levels of all three male categories were significantly higher during the mating season than during the rest of the year. Even though we did not measure levels of corticosterone in the present study, a number of studies confirm the close relationship between stress physiology and haematology (e.g., *Gross & Siegel, 1983*; *Dickens, Earle & Romero, 2009*).

Therefore we suggest that high HCT levels in unpaired individuals during the mating season mirror social stress caused mainly by competition for a mate as well as by constrained access to mates, whereas the haematological differences related to reproductive success might reflect the costs of reproduction in successful geese. Such costs are one of the most significant components of life-history trade-offs and the immune system has been proposed as an important link between reproductive investment and survival (*Sheldon & Verhulst, 1996*; *Deerenberg et al., 1997*; *French, Denardo & Moore, 2007*; *French, Moore & Demas, 2009*; *Harshman & Zera, 2007*; *Cox et al., 2010*).

Although earlier avian studies found higher HCT levels in males than in females (e.g., cormorant, *Phalacrocorax carbo*, *Balasch et al., 1974*; kestrel, *Falcus tinnunculus*, *Rehder, Bird & Laguë, 1982*; sparrowhawk, *Falcus sparverius*, *Rehder & Bird, 1983*), our results are in line with a recent meta-analysis based on 36 published studies did not provide evidence for sex differences in HCT (*Fair, Whitaker & Pearson, 2007*).

Hence, the seasonal differences in relation to pair-bond status we found in our study may reflect differences among greylag geese with different bonding status and energetics (*De Graw, Kern & King, 1979*; *Jenni et al., 2006*). High H/L ratio may reflect physiological stress

produced by competition for partners during the mating season and frequent agonistic interactions among the unpaired individuals, which may not enjoy the stress buffering effect of emotional social support by social allies, notably by a pair partner (*Scheiber et al., 2005*). On the other hand, high H/L ratio may represent the costs of parental commitment for successfully breeding individuals.

Heterophils are the predominant immunological cell type within the Anseriformes (*Lucas & Jamroz, 1961*). They form the first line of cellular defence against invading microbial pathogens (*Maxwell & Robertson, 1998*). Findings in poultry suggest the H/L ratio to be a reliable indicator of social stress (e.g., *Gross & Siegel, 1983*; *Vleck et al., 2000*). This is supported by other studies in birds, which have shown that H/L increases in response to a wide variety of stressors, including long-distance migration (*Owen & Moore, 2006*) and parasitic infection (*Davis, Cook & Altizer, 2004*; *Lobato et al., 2005*; for a review see *Davis, Maney & Maerz, 2008*).

We suggest that the decrease of the percentage of lymphocytes and the increase of heterophils (i.e., high H/L ratio) may also reflect an increase in susceptibility to infections with aging (*Uciechowski & Rink, 2014*), which affects several haematological parameters (*Maxwell et al., 1990*; *Maxwell & Robertson, 1998*; *Prinzinger, Misovic & Nagel, 2012*). In fact, glucocorticoids and oestrogens may reduce T-cell production (i.e., low lymphocytes levels) whereas androgens may increase susceptibility to infections via elevated heterophils levels (*Nelson et al., 2002*). However, as we did not directly measure glucocorticoid hormones, we can only speculate about the relation between haematology and health. Notwithstanding, our results confirm haematology's great potential in studies of ecology and in wild vertebrates, as recently suggested by *Maceda-Veiga et al. (2015)*.

Finally, our results showed that raising condition did affect haematological parameters, suggesting that goose-raised individuals could cope better with the stressful mating season than hand-raised individuals, as the latter showed significantly lower levels of leucocytes as compared to goose-raised geese. Although raising history does not seem to affect other reproductive parameters such as number and weight of eggs laid, hatching success or number of young fledged (*Hemetsberger et al., 2010*), this intrinsic aspect may deserve further attention in the future when considering focal individuals.

In conclusion, our results indicate that the way social and intrinsic factors modulate haematological parameters varied between seasons. Seasonal activities such as reproduction or migration need to fine-tune physiology with weather conditions (e.g., *Dorn et al., 2014*; *Frigerio et al., 2004b*; *Romero, Reed & Wingfield, 2000*). By and large, the investigated haematological parameters varied with individual behavioural investment and stress load. Therefore they may be considered as valid indicators of social burden.

## ACKNOWLEDGEMENTS

We gratefully acknowledge Lara Cibulski, Oliver Elsaesser and Alexander Karl for helping with catching geese in autumn 2012. Blood samples were analysed at the Clinical Pathology Platform of the University of Veterinary Medicine in Vienna (Head Prof. Dr. I Schwendenwein).

### Funding

Financial support was provided by the FWF-Project P21489-B17 (DF, SL), the University of Vienna, the "Verein der Förderer der Konrad Lorenz Forschungsstelle" and the "Herzog von Cumberland Stiftung." The funders had no role in study design, data collection and analysis, decision to publish, or preparation of the manuscript.

### Grant Disclosures

The following grant information was disclosed by the authors:
FWF-Project: P21489-B17.
University of Vienna.
Herzog von Cumberland Stiftung.

### Competing Interests

The authors declare there are no competing interests.

### Author Contributions

- Didone Frigerio conceived and designed the experiments, performed the experiments, analyzed the data, wrote the paper, reviewed drafts of the paper.
- Sonja C. Ludwig conceived and designed the experiments, performed the experiments, reviewed drafts of the paper.
- Josef Hemetsberger conceived and designed the experiments, performed the experiments, contributed reagents/materials/analysis tools, reviewed drafts of the paper.
- Kurt Kotrschal contributed reagents/materials/analysis tools, wrote the paper, reviewed drafts of the paper.
- Claudia A.F. Wascher analyzed the data, wrote the paper, prepared figures and/or tables, reviewed drafts of the paper.

### Animal Ethics

The following information was supplied relating to ethical approvals (i.e., approving body and any reference numbers):

This study complies with all current Austrian laws and regulations concerning the work with wildlife. Catching and blood sampling were performed under Animal Experiment License Nr. 66.006/0010-II/10b/2010 by the Austrian Federal Ministry for Science and Research.

### Data Availability

The raw data has been supplied as Data S1.

### Supplemental Information

Supplemental information for this article can be found online at http://dx.doi.org/10.7717/peerj.2792#supplemental-information.

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
