# Peer review of "Social and environmental factors modulate leucocyte profiles in free-living Greylag geese (Anser anser)"

_PeerJ, doi:10.7717/peerj.2792_

## Round 0.1 · original submission · Major Revisions

Dear Dr. Frigerio,

The two reviews are now in, and your manuscript requires major revisions. Please make the changes requested by the reviewers, or send detailed reasons why you are not following their recommendations. I am particularly concerned with reviewer 1's requests regarding the reporting of sample sizes, reporting statistical power, and checking the sassumptions of the lme analysis. You need to check the following in your linear models analysis, all of which you can do in r: linearity, normality of the residuals (not the original input variables), and homoscedasticity of the residuals. I’m assuming that the assumption of independence is met by your experimental design.

Please follow reviewer 2's instructions regarding reanalysis of your data, because of problems of sample size, and also his demand that you discuss the issue of infection as an alternative explanation for your findings.

Reviewer 1 ·

Basic reporting

This manuscript describes an investigation for the determinants for variability in haemaocrit and differential leukocyte profiles in Graylag geese. The populations compared differed by sex, reproductive status, how they were pair bonded, and how they were reared (human vs goose reared). Blood samples were collected repeatedly from a total of 119 focal birds during three seasons (winter flock, mating season, summer). The authors found significant relationships between haematological parameters and an individual’s social environment as well as reproductive factors. During the mating season, males had significantly higher HCT and lower percentages of lymphocytes than females, while unpaired individuals showed the highest HCT. H/L ratio was positively related to pair-bond status in a seasonally dependent way, with highest values during mating and successful pairs had higher H/L ratio than unsuccessful ones. Also, hand-raised individuals tended to have a higher HCT as compared to goose-raised ones. The authors conclude that HTC, H/L ratio and the proportion of leucocytes are modulated by social factors and conclude that they may be considered valid indicators of individual stress load.

The primarily revision required for this manuscript is additional text on the experimental design and background of the study. The population of birds and how they differ in being raised needs additional description as well as how the results of this study may be used. The objectives or predictions need to be stated clearly in the background.

Experimental design

The experimental design is somewhat straightforward. Although with the number of ways the birds vary with raising, season, pair bounding etc., it not clear the sample size for each group. It is easy to be overwhelmed in such a study with the number of variables and a model selection methodology might be more appropriate. At the minimum, there needs to be the sample size identified for each subgroup or class of variables which could be added to the tables. The graphs also need the sample sizes identified.

Validity of the findings

Once, the sample sizes are identified or there is some language to address statistical power, there can be a better understanding of the validity of the finding. Right, there does not appear to be any fatal flaws since the sampling it somewhat straightforward. However, there is no discussion of the assumptions of mixed linear models such as heteroscedasticity and normality.

Additional comments

Specific Comments to the Authors
Abstract – Line 27 define raising condition in the abstract
Line 62 Define social context – sort of vague and could have different meanings
Line 68 Are there new studies that correlate HCT and testosterone in animals?
Line 76 which = what? Social context or stress responses?
Line 83 Does testosterone increase immediately or over the season?
Line 86 Reproduction costly in most birds. Is there evidence it is even more costly in geese?
Line 90 Parental care is defined as what? And does it change between eggs and after hatching?
Line 111 Individuals differ how? Also, can there be a better description of how birds are hand raised and for how long? Why are they hand raised? When are they released? Are the eggs incubated in the laboratory? Do the adults take to them well after release? More background would be nice on the different raising methods. In addition, why are the birds all supplementally fed? Would you expect different results in your study is the birds were not fed? It seems that the supplemental feeding could alter or dampen the results of the study. This should be addressed in the discussion. What were the birds fed? This should be in the methods.
Line 115 The objectives or predictions need to be stated clearly in the background. What were your predictions you were testing prior to the study from your background information?
Line 136 How young is young of the year? Near hatching or older?
Line 140 Do the birds migrate at all during the year?
Line 145 How were the birds captured? The methods need described.
Line 156 Were the slides fixed?
Line 177 This is where the sample size for each group and subgroup needs to be described. Was it fairly equal for the different groups of raised birds?
Methods – how were geed identified that were not paired? How common is extrapair paternity?
Figure 1 Can the three groups be define again in the legend? Is this both sexes combined?
Figure 2 Is this both sexes combined? What is the sample size for each group?

Reviewer 2 ·

Basic reporting

No comments. Please see my full review in the final box.

Experimental design

No comments. Please see my full review in the final box.

Validity of the findings

No comments. Please see my full review in the final box.

Additional comments

Here is my peer-review of the manuscript by Frigerio et al. This study examines the relative contribution of a number of intrinsic and extrinsic factors (e.g. sex, age, social behaviour) on the blood cell profile of a long-term monitored population of individually tagged Anser anser. The topic is interesting because we need to understand the relative influence of these variables to know the natural variability of blood analyses and hence determine their diagnostic power. However, the low number of replicates in relation to the number of factors examined prevents a reader from having confidence that the conclusions you report are robust. I am aware that current animal welfare standards preclude the use of larger sample size, but the method used by authors is non-destructive. A possible solution is to reduce the number of explanatory factors or the number of interactions considered in the modelling approach. All these factors may be relevant, as well as double and triple interactions, but they need a larger data-set to be tested properly. The degrees of freedom are not provided with stats, but I presume they are low taken into account the number of factors, interactions and the inclusion of a random factor. Statistical analyses therefore require careful attention. Authors should also include the error distribution they have assumed for each dependent variable, and how they have validated modelling outputs.

Regarding the manuscript content, I made some suggestions to improve the clarity (see below). However, I also feel authors should expand discussion on how studies examining white blood cell profiles can differentiate a general stress response from an infectious process. Please see a recent review by Maceda-Veiga et al. (2015 STOTEN Inside the Redbox…). This query should be easy to be addressed by authors as they have determined differential cell count for all 5 white blood cell types. This reviewer also thinks that authors do develop further what they understand by ‘body condition’ and how they have determined this variable. I mean authors directed readers to a paper by Green on mass-length relationships using type II linear regressions, but it is not clear to me how this applies to the current study. Finally, it would be nice to see clearly stated (and potentially better analysed) which intrinsic and extrinsic factors made the largest contribution to each dependent variable. For instance, sex is often not easy to determine in birds and so if the blood variables less influenced by this parameter are highlighted, this may be help other researchers working with this or other similar bird species to select the better set of blood parameters. Hope my comments also will help readers to improve the overall quality of their manuscript.

Specific comments:

H/L ratio and proportion of leukocytes; both are proportions of the same type of blood cell.

Animal’s condition and health – these are two closely related concepts that can be interpreted as the same by some authors. Please be more specific or delete one.

Lines 40 and 46 both refer to the same idea

Line 49 body condition need to be better defined. Do you mean changes in structural size? I guess that taking into account you used Green’s paper later.

Line 52 do you mean cellular or humoral response? Are you referring to innate or acquired immunity? Authors should also explain better why leukocyte numbers are expected to decrease under stress. An infectious disease also triggers a stress response and leads to increased wbc count. Please check Davis et al. (2008 FE The use of leucocyte profiles…) and Maceda-Veiga et al. (2015 STOTEN Inside the Redbox…).

Line 61 Author stated monocytes here but other wbc types are also involved: eosinophils, basophils and heterophils in such a response.

Line 70 I am not convinced that a stress response leads to reduced HCT. It is well-established that any kind of acute stress leads to a significant increase in the number and volume of red blood cells…which also includes rising numbers of immature red blood cells…

---

## Round 0.2 · Minor Revisions

In your revised Ms and the accompanying comments you sent, there are numerical discrepancies that need to be resolved before we can proceed. In addition, I believe you have not properly addressed reviewer #1's request to spell out the sample sizes. That, too, needs to be dealt with before I can go further with this paper. By "proceed" and "go further" I mean send the revised paper back for re-review. Please note that the "Dear Didone" letter is a pre-programmed letter written by the good folks at PeerJ, not by me; this 'comment' is the most important message to you.

Here are the discrepancies I found. In the original paper, you reported having taken data, including 193 blood samples, from 119 geese. In the revised Ms, you report having taken 169 samples from 116 geese. Why the different numbers? Did you delete 24 blood samples and 3 geese? Did you make mistakes? Please explain. In your answer to reviewer #1, you say: "Subset (1) included all 169 samples collected from 105 individuals." In the revised Ms, you say: "A total number of 169 blood samples (N=116 individuals) were collected…" So what is it, 105 or 116 or 119? Where did the N=105 come from?

Reviewer #1 said that it was not clear what the sample sizes were in each group. He/she did not specify the meaning of 'group sample size', but since you had already spelled out the numbers of geese by age, by sex, by social status, and by rearing condition, then surely the reviewer's request was for (a) the only remaining category (phase = summer, autumn, and spring), and (b) combinations of categories (e.g., social status by phase [9 categories]), and (c) blood samples by category (all the categories you give numbers of geese for plus phase). Making up a table for all of the above would be infeasible, as such a table would have 5 dimensions and would contain 3x2x3x2x3 = 108 cells; and, I assume, the experiment was wildly different from balanced (equal number of observations in cells). A reasonable approach would be to give the sample sizes for the categories used to examine interactions via LMMs. To be absolutely clear: First, please add the number of geese sampled after molt, during the autumn, and during the mating season. Second, please indicate the number of blood samples drawn by age, by sex, by social status, by rearing condition, and by phase/season. Third, for combinations of categories that were examined for interaction, please indicate the numbers of geese and blood samples.

---

## Round 0.3 · accepted · Accept

The changes you made answer my concerns.